# Disparities in Accessing Sexual and Reproductive Health Services at the Intersection of Disability and Female Adolescence in Tanzania

**DOI:** 10.3390/ijerph18041657

**Published:** 2021-02-09

**Authors:** Virpi Mesiäislehto, Hisayo Katsui, Richard Sambaiga

**Affiliations:** 1Department of Social Sciences and Philosophy, University of Jyväskylä, 40014 Jyväskylä, Finland; 2Faculty of Social Sciences, University of Helsinki, 00014 Helsinki, Finland; hisayo.katsui@helsinki.fi; 3Department of Sociology and Anthropology, University of Dar es Salaam, 35091 Dar es Salaam, Tanzania; rsambaiga@udsm.ac.tz

**Keywords:** sexual and reproductive health services, adolescence, disability, Tanzania, access to health care, inequality, SRHR

## Abstract

Despite at times having greater needs for sexual and reproductive health (SRH) services, adolescents with disabilities often face challenges when trying to access them. This inaccessibility is further exacerbated during female adolescence. The qualitative study examines how SRH services respond to the characteristics of Tanzanian adolescent females with disabilities. We used the method of empathy-based stories to investigate the perceptions of 136 adolescent females with disabilities of their access to SRH services in Tanzania. The study used thematic content analysis and the Levesque model of health care access was applied as an analytical framework. The results demonstrate that discrimination affects access at different phases of care-seeking, that affectionate behaviour of providers is a central enabler of access, and that for this population access relies on a collective effort. We propose that affection, as an enabler of access, is as an additional provider dimension of access to SRH services for adolescents with disabilities, serving as a “reasonable accommodation” to the health care systems in southern contexts and beyond.

## 1. Introduction

Adolescents with disabilities are as likely to be sexually active as their peers without disabilities and have equal rights to sexual and reproductive health (SRH) services [1]. Adolescence (ages 10–19) is a decisive time in a person’s life during which significant physical, mental, and emotional changes take place [2]. Adolescents with disabilities have unique, and at times greater needs for SRH services but they often face challenges accessing them, especially in many African contexts [3,4,5,6]. These greater SRH needs often result from a lack of information on sexual and reproductive health and rights (SRHR) [4]; sexual abuse and rape, which increases the likelihood of pregnancy [5]; being infected with HIV or sexually transmitted infections (STIs) [7]; and an over-arching stigma [8]. Most of these SRH needs and negative outcomes are exacerbated during female adolescence [3]. Access to SRH services is hindered by physical inaccessibility [9], communication barriers [10], negative attitudes of service providers [4], lack of confidentiality [4,11], costs [4], mistreatment [11], and an overall inadequacy of service delivery [3,4].

The United Republic of Tanzania (Tanzania) has the second youngest population in East Africa [12]. Twelve million of its 54 million citizens are adolescents, an age group expected to reach 30 million by 2050 [13]. Tanzania has one of the highest adolescent fertility rates in the world [14]. Adolescence in Tanzania is associated with a high frequency of child marriage [15], insufficient knowledge about STIs [16], and restricted access to SRH services [17]. Limited access increases the risk of, for example, unplanned pregnancy and STIs among adolescents [18]. What adds to the urgency for SRH services is the socially normalised sexual exploitation of adolescents in Tanzania [19,20].

Of Tanzanian homes, 13.2% have at least one member with a disability [21]. Information on disability remains unreliable, incomprehensive, and incomplete, making the lived realities of persons with disabilities and of different disability sub-groups insufficiently documented [22]. Tanzania has demonstrated commitment to advancing disability rights by ratifying the United Nations Convention on the Rights of Persons with Disabilities (CRPD) and by enacting national disability policies. However, these have not yet translated into notable gains for persons with disabilities. At the policy level, adolescents with disabilities in particular are hidden in the homogenous image of youth and children, and the unique ways in which female adolescence interacts with disability in access to SRH services is neglected [23,24].

The expanding adolescent population, the negative trend in adolescent fertility rates, and the general status of persons with disabilities in Tanzania makes the disparities in the access of adolescent females with disabilities to SRH services a timely and urgent concern. Substantive research has examined the access to SRH services of different sub-groups of adolescents in Tanzania [16,17,25,26,27,28,29]. However, there is a scarcity of disability- and SRHR-related studies from Tanzania [30]. More evidence on SRHR and better access to SRH services among adolescents with disabilities in Tanzania are needed in order to inform health policies and implementation [31,32].

Research has predominantly portrayed girls and women with disabilities in African contexts through vulnerabilities, and little is known about their agency in terms of SRHR [33,34]. Greater engagement of adolescent females with disabilities with their SRH can be positively associated with the utilisation and effectiveness of services [3], and thus the factors influencing their agency need to be better understood. Consequently, this study employed a definition of access to health care that sees access as a result of “the interface between the characteristics of persons, households, social and physical environments and the characteristics of health systems’ organisations and providers” [35] (p. 6). This broad understanding of access concurs with the CRPD definition that sees disability as an “interaction between persons with impairments and attitudinal and environmental barriers that hinders their full and effective participation in society on an equal basis with others” [36].

Recently, the Levesque health care access model [35] has proven beneficial in research on access to SRH services for persons with disabilities [37]. The model recognises the agency and capabilities of socially marginalised people rather than reinforcing a view of them as passive recipients [38,39]. It also allows looking at the variations in the access of different populations, to enable a better understanding of health care disparities [35]. The framework entails the relevant access-related concerns of persons with disabilities, such as accommodation and appropriateness [36,40]. The model captures five service provider dimensions: (1) Approachability, (2) Acceptability, (3) Availability and Accommodation, (4) Affordability, and (5) Appropriateness; and five service user dimensions: (1) Ability to perceive, (2) Ability to seek, (3) Ability to reach, (4) Ability to pay, and (5) Ability to engage. This article uses the analytical framework of the Levesque model of health care access to examine how different dimensions generate service users’ access to SRH services or their disparity in accessing them. The objective of this study was to increase the understanding of access to SRH services of adolescent females with disabilities in Tanzania. The study aimed to answer the following research questions: How adolescent females with disabilities perceive the accessibility of SRH services? What characterises their capabilities to access SRH services?

## 2. Materials and Methods

A qualitative participatory research methodology was utilised to explore access to SRH services. The participants were 136 female Tanzanian adolescents with disabilities, aged 10 to 19. The average age of the participants was 15.5 years. The majority of the participants had hearing impairments (*n* = 42); one of them also had a physical disability. Others had physical impairment(s) (*n* = 33) and albinism (*n* = 32), and of these some (*n* = 11) also had partial visual impairments. In addition, the participants had intellectual impairments (*n* = 13), visual impairments (*n* = 12), and partial visual impairments (*n* = 4). Purposive sampling was used to identify the participants from a sampling frame established together with the Tanzania Federation of Disabled People’s Organizations. Research was conducted in 13 locations across three regions in mainland Tanzania, mostly in school settings.

This research draws from data consisting of 257 empathy-based stories (EBS) that were on average 902 characters long, produced by the 136 participants. EBS are fictional short stories narrated by study participants as a response to a frame story [41,42]. The verbally collected stories were transcribed verbatim by the interviewer immediately after the interviews. The transcripts were translated from Kiswahili into English and cross-checked by a member of the research advisory committee and the researcher to ensure the quality of the data. This committee consisted of members of the Tanzania Federation of Disabled People’s Organizations to ensure that the persons with disabilities played an active role in the study. The research advisory committee was pivotal in developing a culturally appropriate methodology, providing insights at the analysis phase and engaging in discussion on the results. In addition, the research advisory committee was instrumental for ensuring the validity of the study. By triangulating data from multiple sources, namely, empathy-based stories and research advisory committee proceedings, the in-depth interpretation of the data was enhanced.

The method of empathy-based stories (MEBS) was employed to acquire perspectives rooted in the participants’ narratives. MEBS is grounded in constructivist and relativist epistemologies, where reality is understood as contextually and socially constructed [41]. MEBS assumes that the reality and narrative both reflect and imitate each other and the ability of the narratives to reflect personal meanings is more important than capturing lived experiences [43]. Thus, the focus in MEBS is not on researching lived experiences, but perceptions. MEBS is considered an ethically sensitive data collection method, as it allows participants to distance themselves from controversial and sensitive topics [43]. In addition, although the frame stories evoke perspective-taking that may lead to the activation of participants’ personal experiences, they may decide whether or not they wish to disclose these.

EBS were narrated by the participants as a response to the short fictional frame stories provided by the researcher. The frame stories, as well as the visual and audio aids accompanying them, were developed with the research advisory committee and piloted prior to the data collection. Considering the unique characteristics of the participants, and their educational status, the participants took part in the storytelling verbally or in writing. The verbal storytelling sessions resembled individual interviews, in which the participants first responded to the frame story. After this it was possible to acquire more insights regarding their views through predetermined prompt questions. All participants were given the opportunity to respond to both frame stories. It would have been ethically problematic if some participants only responded to the negative variation of the frame story. The verbal data collection sessions were conducted in Kiswahili and in Tanzanian sign language.

The frame stories were constructed around two protagonists: Fatuma and Nuru. They reflected two common issues that require adolescent females to seek SRH services in Tanzania [29]. The frame story variation was constructed to simulate perceptions on the accessibility and inaccessibility of SRH services. The stories utilised were as follows:


*Fatuma John is an 18-year-old girl with disabilities and she is pregnant. She needs to attend the health clinic. After visiting the clinic, she feels happy and safe. Tell us what happened at the clinic that made her feel happy.*



*Nuru Hassan is a 15-year-old girl with disabilities. Nuru has a disease in her private parts and she is experiencing discomfort. She needs to see the doctor. In the meeting with the health professional something goes terribly wrong. Afterwards Nuru is really upset. Tell us what happened when she met the health professional. Tell us what disturbed her so much.*


Ethical considerations informed the design and methodology of the study. This was especially important as the studied adolescents are often made vulnerable by multiple discrimination in the society. Ethical concerns were also related to the culturally sensitive topic of SRH services. Accessibility to the locations and the methods and materials were carefully considered. Efforts were made to create safe research spaces that would ensure and promote physical and psychological safety as well as freedom of expression. Information sessions were held with participants and their guardians on the research sites. These sessions included information about the research aims and the storytelling sessions, confidentiality and anonymity, and the right to decline and withdraw from the study at any time. After ensuring a genuinely informed consent through an “explain-back” protocol, a written consent was received from participants and their guardians. For those that are illiterate, or with intellectual, visual, or hearing impairments, information and consent procedures were made accessible by using visual aids and by providing an option to give consent through an audio recording or by using a thumbprint. Processual view of consent was maintained throughout the study, and the participants were reminded in the beginning of their storytelling sessions about the voluntary nature of the study and the right to withdraw from the study at any time. Body language and ease of the participants was observed and documented. A storytelling session would be ended in case a participant portrayed discomfort. Ethical clearance for this study was obtained from the National Institute of Medical Research in Tanzania

The study used data-driven thematic content analysis. The researcher first closely examined the notions of accessibility and inaccessibility within the data by separating the story variations. Then recurrent patterns and topics were freely coded. The codes began to resemble dimensions of access in the Levesque model of health care access [35]. The appropriateness of the dimensions was discussed with the research advisory committee, as were the proposed codes under each dimension. Based on this triangulation of data with the research advisory committee, the thematic locations of the codes were refined and thematised under the Levesque model’s dimensions of access.

## 3. Results

The findings demonstrated that, according to the participants’ perspectives, the SRH services did not respond well to the characteristics of the adolescent females with disabilities in terms of Approachability, Acceptability, Availability and Accommodation, Affordability, and Appropriateness. Moreover, the provider characteristics had adverse effects on the service users’ abilities to access services. In this chapter, we first elaborate on supportive networks and access, second on the social acceptability of the service users, third on the interpersonal characteristics of the providers, and finally on violence inside and outside the health facilities.

### 3.1. “Go and Bring Your Mother, So We Can Talk to Her”: Supportive Networks and Access

The ability to seek health care is often associated with personal autonomy regarding seeking care and the capacity to choose health care options [35]. The findings of this study demonstrated that theorisation regarding access that emphasises self-determination is not always compatible with the lived realities of the studied adolescents in the Tanzanian context. Overall, the narratives did not present independence and self-sufficiency in a positive light. When such independence was demonstrated, it resulted in poor service-seeking outcomes. The narratives showed that for adolescent females with disabilities, seeking care is not an individual’s decision; it is influenced by relationships and collective decision-making. The participants’ stories also depicted a gap in the reasonable accommodation of services regarding their disability. For instance, there were no mention of formal assistance provided at the health facilities. Much of this gap was filled by relying on support networks, concurring with previous literature [4]. The role of supportive relationships has previously been associated with positive health care outcomes for persons with disabilities [44]. Various relationships can determine the ability to seek care, hence either restricting or enhancing access to care. Such relationships in this study were those between the studied adolescents and family members, teachers, matrons, community members (e.g., neighbours), other patients, and health care professionals. Good relationships with the surrounding community and family members were perceived to enable access. Support was largely described as enabling, but as it was based on benevolence and not fully reliable, as demonstrated below:


*I told them [my parents] I can’t go to the hospital alone. I told my dad and mum and they said ‘Aah, we will take you later’. I decided to go to the nurse, I explained the situation to her, I asked her to help me. […] She looked for the medicine and gave it to me. She helped me and I went home. So, she listened to me and helped me. I fell asleep but kept thinking ‘Why am I alone? I have problems but no one to help me’. I met with my friend and told her. I met with another one and asked her to help me. So, she is now helping me.*
Participant with hearing impairment, aged 19

There was a strong reliance on supportive networks in terms of finances, mobility support, communication, and safeguarding. Generally, the narratives portrayed households with low incomes, and health insurance was considered very rare. A collective effort was needed to mobilise funds to access SRH services, a process which was not always streamlined. Inability to pay was related to the autonomy and decision-making linked with choosing to seek care and make decisions about one’s care. Although parents were usually responsible for health care payments, it was common that financial resources needed to be mobilised within the household and the community. Mobilising financial resources was at times portrayed as a time- and energy-consuming responsibility of the adolescent. The narratives also depicted that despite various efforts, sometimes they were in vain, and thus services or parts of the service were denied.

For those with physical and visual impairments, the main reason for needing an escort was the physical journey. For the studied adolescents with albinism, the need for assistance was related to superstitious beliefs regarding violence against persons with albinism. The service users needed support to reach the facilities, and the service providers often did not respond well to their ways of communication. Self-expression was hindered by a lack of the right terminology to explain, feelings of shame, and Kiswahili- or sign language-related barriers. Communication was not only a challenge for those with hearing impairments.

According to Burke et al. [4], the most common way to overcome communication barriers was by employing the assistance of a family member or using writing as an alternative means of communication. However, being accompanied by assistants and using writing were at times seen as problematic. The former jeopardised the privacy of the service users and the latter excluded those who were illiterate.


*She [Nuru] went to the doctor but no one took care of her. When she tried to explain her need, nobody understood her. They were, like, ‘aha!’. They examined her but they were, like, ‘Hmmm? We can’t help you with this, this is very serious’. So, she was afraid, very ashamed. So, the doctor told her ‘Go and bring your mother, so we can talk to her’. Because of the way they were mistreating her she went to get her mother and explained to her: ‘Mother please forgive me, I have this infection and it is not good at all, I am in a lot of pain. Let’s go to the hospital’. Her mother helped her and went to the hospital. She went to the hospital to find the doctor who was mistreating her.*
Participant with hearing impairment, aged 16

Arranging assistance was not always effortless. Shame related to the need for SRH care was sometimes perceived as the major barrier to requesting support. For those who were students, an escort was often provided by the school but for out-of-school adolescents it was disproportionately more difficult to arrange an escort. When assistance was not available, it was common to remain at home, which delayed access or made it altogether impossible.


*[…] just because someone has a disability it does not mean that there are people who are always ready to accompany them all the time. You find a girl with a physical disability crawling on the floor, she starts to think about how to get out of the house and get to the hospital, that is more than a challenge. When she starts thinking of someone to take her to the hospital, she just gives up. Now it depends on one’s illness, what it is like or when she got it because there is not always someone there to help you when you are in trouble. So, the issue of distance is a challenge and also finding someone to escort you.*
Research advisory committee member with physical disabilities

Although supporting relationships are an essential enabler of access, previous studies have shown that they often compromise the privacy and confidentiality of the service users [40]. SRH needs were portrayed by the participants as a private issue; however, sometimes they had to rely on whoever was available to accompany them. Reliance on financial assistance in particular was perceived to challenge confidentiality and privacy.


*For girls like these, they have no other place to get money, so she will have to tell her parents because she has no money. She can’t do anything at all. For example, for those of us who are blind, it is difficult, because in everything that you do you need a helper, hence there will be someone who’ll know what is going on even if you have your own money. The issue here is how will you go there? There is no way to hide. Even if you have an abortion you can’t say the secret has remained between you and the doctor. Never! So for us there is absolutely no way to hide.*
Research advisory committee member with visual impairment

### 3.2. “It Would Be a Great Idea If God Took Me”: Social Acceptability of Service Users

The findings concurred with previous research on the general adolescent population in Tanzania, in that the participants did not perceive themselves as clients of SRH services, as if they were socialised to believe that the services were only for those who are pregnant and married [16]. Some narratives also portrayed uncertainty of whether a girl with disabilities could or should have children or could get married at all, which would restrict their access even further and negatively affect their social status in the future. Firstly, various social and cultural factors determined the perceived appropriateness of the studied adolescents seeking care. Secondly, social and cultural factors related to sex and pregnancy influenced how the participants accepted features of the SRH services.

SRH service acceptability highlighted the intersecting social categories that discriminate adolescent females with disabilities in complex ways when seeking access. In the participants’ narratives, disability intersected with the following social categories: age, pregnancy, marital status, educational status, and impairment type. For instance, when disability compounded with age and pregnancy, further discriminatory aspects were demonstrated by the participants:


*A normal person is warmly welcomed, they even help with handbags but if they see a pregnant person with disabilities, they start giving her unsolicited advice. They give her medicine then tell her ‘You can just go home’. So, she goes home, takes the medicine and waits for her due date, as it is still early for delivery. Other persons with disabilities feel like they are wasting time going back and forth to the hospital. This is because first and foremost she is pregnant, as well as a person with disabilities, and for this reason it is not good to go to the hospital earlier than on her due date.*
Participant with hearing impairment, aged 19

Compounded by the above, a certain impairment type may have a negative effect on how the studied adolescents are perceived as SRH care-seekers. When a hearing impairment or albinism co-existed with adolescence, acceptance of the adolescent was especially hindered. In contrast, being married and/or having at least a basic level of education increased the acceptability of disability.

The service providers, family members, and even other patients were engaged in this discriminatory behaviour.


*When seated at the bench with other patients, they might start talking badly about me saying things like ‘Look at this child! She’s an albino. She’s disabled. Look at how her eyes stick out!’ […] Other patients might say ‘Look at those eyes! They’re bulging!’ So, they can say these things that make me feel bad and make me not want to go to the hospital. I find myself in despair and start thinking it would be a great idea if God took me.*
Participant with albinism and visual impairment, aged 12

The participants’ narratives demonstrated that it is perceived as forbidden for adolescent females with disabilities to engage in sexual activities. These negative and judgmental attitudes towards sexual activity influenced how acceptable it was for them to seek SRH services. The consequences of risky sexual behaviour, pregnancy, and STIs were also highly condemned by the girls themselves and their communities. This judgmental attitude towards sexual activity was presented as a protective mechanism placed on the studied adolescents by their communities. As such, it was partly accepted by the studied adolescents. However, it was also perceived as a limitation to their relationships, to the extent that they were instructed to avoid any social contact with boys and men. Unlike many studies that highlight how the surrounding society assumes persons with disabilities to be asexual [33], the participants’ narratives included no such notions. On the contrary, the narratives embedded a view that they were perceived by their communities to be sexually active because of their disability and age.

If pregnancy were perceived as socially acceptable, seeking services would also be perceived acceptable. However, the social acceptance of pregnancy was determined by marital status, age, disability type, education, frequency of pregnancy, poverty, and how the pregnancy was initiated (e.g., in a relationship or through rape). Marriage, higher age, education, and infrequent pregnancy were factors that increased the studied adolescents’ acceptability to seek SRH services. Regardless of many external factors, some of the adolescents’ narratives still indicated an internalised view of themselves as unacceptable clients.

A participant with a hearing impairment, aged 14, gives advice to Fatuma. Her account illustrates how advantage and disadvantage co-occurs with pregnancy and disability. Such dynamics were prevalent in the narratives.


*I would tell her if that is the case [being pregnant] then there’s nothing you can do about it. Because if she aborts, she could also die. She can kill the unborn baby, but she can also die. I would advise her to persevere until she gives birth as she’s not alone in this. There are others who are also persons with disabilities and pregnant. She’s lucky because she’s pregnant and is still able to go to the hospital. There are others who are too afraid and stay home.*
Participant with hearing impairment, aged 14

### 3.3. “They Should Show Love, Caring and Kindness”: Interpersonal Characteristics of the Providers

Shame, among other negative emotions regarding the need for health care, was portrayed as restricting the ability to seek care. The interpersonal qualities of the providers did not respond well to such negative emotions. Their attitudes towards the studied adolescents were reflected in patient blaming, neglect, refusing to treat, laughing at, verbally insulting, yelling, and openly despising the condition of the person or their disability. These findings are aligned with those of Burke et al. [4], which indicated that poor service provider attitudes are barriers to SRH access among young persons with disabilities.

The narratives highlighted the absence of being attended to as person first, rather than disability first. The impairments were shown to supersede the need for SRH. The participants described how service providers perceived disability as a strong label even when it shadowed the actual service needs.


*When you go to the doctor, people with disabilities are despised. So that everyone is comfortable, they should all be treated equally so as to avoid asking ‘Doesn’t she hear at all?’ ‘Can’t she speak at all?’. Doctors should give medical care according to the patients’ needs.*
Participant with hearing impairment, aged 17

In addition, the SRH condition of the service user could also become such a strong label that she is no longer treated person first but as the manifestation of an illness or a health condition.

The participants vividly narrated what would be the ideal characteristics of an SRH service provider. Demonstrating affection was the most prevalent description.


*As a person with a disability, my advice to doctors is [that] they should know that when you scold and criticise a person with disabilities, she won’t be at peace. Therefore, they should show love, caring and kindness.*
Participant with physical disabilities, aged 15

Demonstrating affection included giving a warm welcome, using kind and encouraging words, giving positive feedback, having the ability to listen and explain well, taking affirmative action towards persons with disabilities, and assuring the continuity of the services.

### 3.4. “I Don’t Want Your Money; I Want Something Else”: Violence within and Outside Health Facilities

The findings demonstrate that SRH services appear to reflect the social dynamics prevalent in society, including sexual violence and exploitation of adolescents and women with disabilities [20,45,46,47]. The narratives entailed gross depictions of the studied population exposed to transactional sex and sexual exploitation as a result of the aforementioned financial dependency and the desire for privacy in SRH services. The narratives described how demanding sexual favours in exchange for care was embedded in society.


*There is financial corruption and sexual corruption. […] So, I advise him [the doctor] to avoid such things. Should he engage in such activities he will be violating his work ethics and also disrespecting himself.*
Participant with albinism, aged 14


*The specialist told her: ‘The cost of my help is too high, I don’t know if you can manage it’. Nuru told him: ‘If it’s the money you want, just tell me how much so that I can ask for help from my relatives and friends’. The doctor replied, ‘I don’t want your money, I want something else’. Then the girl was devastated and started crying.*
Participant with physical disabilities, aged 15

There is evidence that sexual violence by health providers occurs worldwide [47] yet a secrecy surrounds sexual misconduct in medicine, making it difficult to identify cases [48]. These findings concur with other studies, which indicate that, in SRH services, for those who are perceived as deviant of social norms, abuse is more common [47]. The findings are also in line with previous literature, in that the female gender [48] and considerable power disparities between the health provider and the service user may combine to generate sexual violence in health care settings [49]. Here, adolescent females with disabilities were seen as not conforming to the social norm of an SRH service user due to their disabilities, age, and other intersecting factors, which could be contributing factors to sexual violence. The narratives included notions of a lack of confidence when in the company of the service provider, which may denote a power distance between the adolescent females with disabilities and the health care providers. Female health providers were preferred over males due to the fear of males requesting sexual favours and even of sexual violence.


*Participant: She the [female doctor] will remove your clothes and I don’t want a man to see me naked. […] Male doctors have really bad behaviour.*



*Researcher: What kind of bad behaviour do they have?*



*Participant: Having sex with patients.*
Participant with intellectual impairment, aged 19

There were references to sexual violence and exploitation by health providers even without association with payment. Depictions of sexual advances indicated the presence of sexual violence among the participants’ lived realities within and outside health care settings.


*There are some doctors who are just crazy. You go to the hospital expecting to be treated but he starts doing other things instead of treating you. He starts telling you strange things. You get out of there angry and frustrated because you went there to seek medical treatment and not to be asked strange, silly questions or to be seduced.*
Participant with hearing impairment, aged 14

In addition to the violence within health care settings influencing access, the ability to engage in one’s care is also further restricted by sexual violence towards the studied participants within society. The evidence that violence prevents engagement in health care by persons with disabilities is consistent with findings of other studies [50].

## 4. Discussion

The unique role of relationships in the disparities of SRH access of adolescent females with disabilities in Tanzania cuts across the principal findings. Here we further discuss the role of relationships, first by looking at the relationship between the service user and provider, and secondly, by focusing on patients as members of collective entities. Finally, we elaborate on the relationship between the service providers and their environments. From there, we move on to the role of affection in these relationships.

A negative relationship with service providers makes SRH services an extension to the stigma, neglect, and abuse so often faced outside health facilities by the studied adolescents [20,45]. SRH service access disparities were perceived to be generated by the unprofessional behaviour and discriminating attitudes of the service providers towards the studied adolescents. SRH services’ failure to treat patients respectfully concurs with other studies from Tanzania [51,52]. Disability embodiment appeared to further reinforce disrespect. Although the service users portrayed high expectations of psychosocial support from SRH services, the services in turn were perceived to offer them neglect, insults, and abuse, often leading to a perceived countereffect on their well-being. There is strong evidence that affectionate communication increases positive health care outcomes [53], and good interpersonal skills have proven to positively influence the utilisation of services [54]. Thus, the service providers play a pivotal role in increasing the quality of care and positive health outcomes. Their interpersonal skills could perhaps be developed without an undue burden on the Tanzanian health system by adopting simple techniques of affectionate communication into their encounters with service users. This is indeed good news for the Tanzanian health care system, whose resources are scarce and other aspects of SRH services may be costly and hard to change. In the relationship between the service users and providers, disturbingly, sexual violence appeared to negatively influence access. Considering the limited opportunities of the adolescents to express themselves in the study, their epistemic standpoint becomes important when drawing conclusions about sexual violence in the context of SRH services. The often-silenced views of this population need to be listened to, taken seriously, and further explored to overcome the violence and overall disparities in SRH care access.

The current study has presented adolescent females with disabilities as part of collective entities: families, communities, and informal networks of care. It shows that the family and community largely determine SRH access and that these relationships are in turn determinants of access disparities. This collective approach to disability, which sees disability as a family or a community concern rather than an individual’s concern, is still marginalised in disability theorisations [55]. Nevertheless, Aldersay [56] has developed a Tanzanian approach to family and disability, which argues that building on existing strengths of families and communities is a more sustainable intervention strategy than focusing on their deficits. This strength-based view can be expanded further, as the findings support the understanding that in southern contexts “charity is a way of survival when no institutionalised support is available” [57] (p. 135). Traditionally, the charity approach and human rights discourse have been contradictory. This juxtaposition has been perceived problematic in disability research in the global South [57,58,59] as, for example, the well-being of persons with disabilities is largely dependent on charity [57]. This tension between the approaches could be used to recognise a continuum between charity and rights approaches. Such non-binary thinking is certainly a lesson to learn from the southern perspectives on disability [60,61].

Health care often reflects the social dynamics that are normalised in society overall [47], including the marginalisation of persons with disabilities and their families. To understand the SRH service providers amidst social forces, we need to turn our attention to the drivers of the injustices. Despite policy-level commitment to adolescent-friendly SRH services [26,62], the facilities that actually implement it are few [29]. The problematic implementation of adolescent SRH policies may result from the complex discursive SRHR landscape that spreads across the restrictive–liberal divide in Tanzania [63]. According to Bylund et al. [28], complexity in adolescent SRHR may be intensified by contradictory and inconsistent messages regarding SRHR policies, the legal framework for providing services to underaged adolescents; the president’s personal views on SRHR; and community attitudes. The inequalities in the Tanzanian SRH services appear to be rooted into ageism, ableism, and patriarchy, which, according to the results of this study, intersect with disability and female adolescence, but also with marital status, poverty, STI status, social networks, and educational status. These intersections produce specific circumstances of power, disadvantage, and identity that differ from the lived realities of other adolescents. In accordance with the present findings, previous studies have also pointed out that SRH service providers navigate within a complex gendered environment, a moral framework in terms of premarital sexual activity, socio-cultural norms around marriage [64,65], and the diverse characteristics of service users. This ambiguous social and political environment provides SRH service providers with inadequate response mechanisms to the characteristics of the studied adolescents.

Affection is an overarching theme across SRHR research and within the relationships of persons with disabilities [57,66,67]. According to Shakespeare [68] (p. 3), “rights alone are not enough to promote the well-being of disabled people, and that charity—defined broadly as love and solidarity—must also play an important part”. Charity, reconceptualised here as affection, is an indispensable part of disability in southern contexts in which the well-being of persons with disabilities is much dependent on it [57]. In the context of CRPD, reasonable accommodation, and access to SRH services, affection can also be understood as a pathway of “necessary and appropriate modification and adjustments”, which will unlikely become an “undue burden” to systems, collective entities, and individuals [36]. As proposed by Katsui and Mesiäislehto [69], affection is a central aspect of the well-being of persons with disabilities and should be included as a principle in global disability inclusion. The present study raises the possibility that affection is also a central enabler of access to SRH services and is thus proposed as an additional provider dimension for the SRH services of adolescent females with disabilities in southern contexts and beyond.

This study had various limitations. The ethical and methodological choices of utilising MEBS means that the findings are based on perceptions rather than on real-life experiences. Recruiting participants mostly from school settings through an organisation of persons with disabilities may have resulted into exclusion of adolescent females with disabilities in the furthermost margins of the society. The participants may have shown social desirability bias when discussing sensitive and at times socially unacceptable behaviours and may not have narrated the stories without self-censorship. Thematically, from the broad scope of SRH services, the study was confined to exploring access to services related to adolescent pregnancy and STIs. The participatory and storytelling approaches provided an opportunity to minimise the power differences between the researcher and the participants. The triangulation of the results to the research advisory committee was an attempt to reduce any biases caused by the researcher’s background and to strengthen the validity of the study.

## 5. Conclusions

This study is among the first to examine the inequalities faced by adolescent females with disabilities when attempting to access SRH services in Tanzania. The results show that the studied adolescents are capable of more engagement in their SRH service access, but that the characteristics of the providers and systems do not respond well to their characteristics. The findings of this study broadly support previous work that demonstrates the challenges when accessing SRH services [3,4,5,6], such as physical inaccessibility [9], communication barriers [10], negative attitudes of service providers [4], issues of confidentiality [4,11], and mistreatment [11]. Although this consistency with previous studies is an important achievement, the study also brought forth new insights into the intersection of disability and female adolescence with access to SRH services. The main findings were that various identities simultaneously produce advantages and disadvantages and that discrimination affects the access of these adolescents to SRH services at different phases of care-seeking. Moreover, the study revealed that service providers’ affectionate interpersonal skills are a central enabler of access, and that access is a collective effort. It also showed that SRH services reflect the social dynamics of a society, including sexual violence against adolescent females with disabilities.

The findings of this study have several implications. In Tanzania, SRH-related policies and health care programmes should address the characteristics of this population and consider affection a reasonable accommodation to ensure equal access for adolescent females with disabilities to SRH services. These findings should be incorporated into the national adolescent-friendly SRH service response, as well as the National Plan of Action to End Violence against Women and Children. Safeguarding principles within health care should be emphasised, including transitions within, to, and from the facilities. Further research based on the experiences of this population is urgently needed to develop a full picture of sexual violence inside and outside health facilities, which can inform prevention. To contribute to this, the first author is in the process of writing a publication on sexual violence against the studied population. There is also abundant room for further progress in the methodologies used in southern disability research on the SRHR of adolescents. Future publications will use experiences from this study to elaborate on this.

## Data Availability

The data of this study are not publicly available due to ethical restrictions and because they contain information that could compromise the privacy of the study participants. The data presented are available on reasonable request from the corresponding author.

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
