# Peer review of "Disparities in Accessing Sexual and Reproductive Health Services at the Intersection of Disability and Female Adolescence in Tanzania"

_ijerph, 2021, doi:10.3390/ijerph18041657_

Round 1

Reviewer 1 Report

The manuscript presented is very interesting and of enormous relevance. It is worth paying attention to the difficulties that adolescent women with disabilities have in accessing sexual and reproductive health services.

On the other hand, I must congratulate the authors for using a qualitative methodology that allows for the evaluation of access to health services. The effort to take an intersectional look at a phenomenon such as disability, gender, and poverty situations must also be recognized.

The introduction is very appropriate and complete. It is important to highlight the positive view that the authors have about sexuality in people with functional diversity. As well as the results and the discussion.

The following are a series of recommendations to improve the manuscript presented:
- In the materials and methods it is important to reference the methods used.
- Similarly, it would be necessary to specify how participants were informed of the conditions of participation and how informed consent for their participation was obtained.
- It is also not clear how the stories of the adolescents were collected. It would be interesting to expand on this topic.
- It would also be interesting, given the novelty of the empathy-based story method, for the authors to expand on how they have used this approach in analyzing the adolescents' stories.

Reviewer 2 Report

Thank you for the opportunity to review this interesting and novel paper, which uses an innovative approach to explore issues of access to SRH services among adolescents with disabilities in Tanzania.

I have some major, and also minor, concerns.

  1. I do not understand why a Tanzanian author was not included, given that the conduct of the research was in Tanzania in the local languages. For me, this is a major issue and the paper should not be accepted until it is resolved. I apologise if I have misunderstood the origin of the authors.

  1. I was interested in the MEBS theory and how it could be used to uncover sensitive information. However, the authors need to be far more cautious about the interpretation of the information. As I understand, this will give an indication of what adolescents with disabilities perceive to commonly happen for people with disabilities. It does not show their actual experience, and may be influenced both by their own stereotypes and prejudices as well as social desirability bias (as the authors note). The discussion needs to be radically rewritten, therefore, to avoid over-interpretation. Here are examples of where this over-interpretation was done:
  • Line 423 – “SRH service access disparities are generated by the unprofessional behaviour and discriminating attitudes of the service providers towards the studied adolescents.”
  • Line 427 – “Although the service users had high expectations of psychosocial support from SRH services, the services in turn offered them neglect, insults and abuse, often leading to a countereffect on their well-being.”

  1. The authors present the Levesque framework, but do not use this approach to interpret their findings. It would be helpful if reference could be made in the results and the discussion to this conceptualization.

  1. I am concerned about the focus on promoting “affection” and “affectionate behaviour”. Healthcare professionals should not be charitably minded towards adolescents with disabilities, but should respect their right to SRH and have the skills and support to serve their needs. I am also not comfortable with the promotion of the charitable approach in the paper. I suggest that all discussion on “affection” and charity be removed.

  1. The participants included people from age 10, and also people with hearing and other impairments. Information needs to be given on obtaining informed consent, particularly for these groups.

  1. The paper is currently very long, and could be reduced.

Minor comments:

  1. Abstract
  • More information should be given on methods – how many people were interviewed, how was data collected (describe MEBS), who were the participants, how was data analysed?
  • Remove mention of “Affectionate behaviour”.

  1. Introduction
  • The introduction is generally well written and well referenced
  • The authors could also mention that adolescence is also a time when health and healthcare seeking patterns are established
  • This sentence is not clear “Most of these SRH needs and negative outcomes are exacerbated during female adolescence [3].” as the whole focus of the paper is on adolescents. Do they mean that this is more pronounced in females than males adolescents with disabilities?
  • Line 33 – access to SRH for all adolescents, or adolescents with disabilities?
  • The first paragraph of the introduction focuses on the African literature, which is fine, but needs to be explicitly stated

Methods

  • More information is needed to justify the use of MEBS, and whether there is evidence that this approach captures real concerns of individuals
  • What adaptations were used to include people with disabilities
  • Who were the interviewers and what was the training?

Results

  • As mentioned already, the presentation of results does not relate to the Levesque framework. It would be helpful if this framework could be brought in to help structure the findings.
  • I believe that there is over-interpretation of the findings. For instance the statement: “The findings demonstrated that the SRH services did not respond well to the characteristics of the adolescent females with disabilities in terms of Approachability, Acceptability, Availability and Accommodation, Affordability, and Appropriateness.” Is going beyond what information is captured in MEBS, as far as I can see.
  • It is not clear to me why there are there so many quotes from the research advisory committee?
  • Intersectionality is not well covered. – Line 269 – “In the participants’ narratives, disability intersected with the following social categories: age, pregnancy, marital status, educational status, and impairment type.” Information was not presented to support this statement.
  • Results should only present findings from this study – the comparison with the literature should be in the discussion

Discussion

  • To reiterate, I am concerned in the discussion about 1) over-interpretation of the results and the authors need to be more cautious about what their findings show, 2) advocating for charity and/or affection.
  • In terms of implications, I absolutely do not agree that healthcare professionals need to be trained on affection. They should be trained on the rights of people with disabilities. Line 525
  • The authors are also making this recommendation without evidence of effectiveness – line 432 - Their role could be developed without undue burden on the health professionals’ resources by adopting simple techniques of affectionate communication.
  • Line 470 – no evidence presented to support this statement “As demonstrated in the current study, in the Tanzanian SRH services, ageism, ableism and patriarchy compound with disability, age and gender, but also with marital status, poverty, STI status, social networks, and educational status. ” Moreover, only females and disabled people were interviewed
  • Of course people are in families, but marginalisation of disability is in part because of lack of voice of individuals so working on families to improve SRH of disabled people is problematic.
  • The authors could also mention as limitations that they identified people through DPOs and mostly in school settings, so there would be bias in the sample. Only disabled people were interviewed and so comparison is impossible. Furthermore, the MEBS does not capture direct experience.
  • The discussion is currently quite long and could be reduced.

Round 2

Reviewer 1 Report

The modifications made by the authors correspond to the requirements of this reviewer.